# The Antitumoral/Antimetastatic Action of the Flavonoid Brachydin A in Metastatic Prostate Tumor Spheroids In Vitro Is Mediated by (Parthanatos) PARP-Related Cell Death

**DOI:** 10.3390/pharmaceutics14050963

**Published:** 2022-04-29

**Authors:** Diego Luis Ribeiro, Katiuska Tuttis, Larissa Cristina Bastos de Oliveira, Juliana Mara Serpeloni, Izabela Natalia Faria Gomes, André van Helvoort Lengert, Cláudia Quintino da Rocha, Rui Manuel Reis, Ilce Mara de Syllos Cólus, Lusânia Maria Greggi Antunes

**Affiliations:** 1Departament of Genetics, Ribeirão Preto Medical School, University of São Paulo, Ribeirão Preto 14049-900, Brazil; kattuttis@gmail.com; 2Department of General Biology, Center for Biological Sciences, State University of Londrina, Londrina 86057-970, Brazil; larissa.oliveira@uel.br (L.C.B.d.O.); julianaserpeloni@uel.br (J.M.S.); ilcecolus@gmail.com (I.M.d.S.C.); 3Molecular Oncology Research Center, Barretos Cancer Hospital, Barretos 14784-400, Brazil; izabela.faria.tk@hotmail.com (I.N.F.G.); ahlengert@gmail.com (A.v.H.L.); ruireis.hcb@gmail.com (R.M.R.); 4Department of Chemistry, Center for Exact Sciences and Technology, Federal University of Maranhão, São Luís 65080-805, Brazil; claudiarocha3@yahoo.com.br; 5ICVS/3B’s—PT Government Associate Laboratory, 4806-909 Guimarães, Portugal; 6Life and Health Sciences Research Institute (ICVS), School of Medicine, University of Minho, 4710-057 Braga, Portugal; 7Department of Clinical Analysis, Toxicology, and Food Sciences, School of Pharmaceutical Sciences of Ribeirão Preto, University of São Paulo, Ribeirão Preto 14040-903, Brazil; lusania@fcfrp.usp.br

**Keywords:** *Fridericia platyphylla*, prostate cancer, chemotherapy, tumor spheroids, cell death

## Abstract

Metastatic prostate cancer (mPCa) is resistant to several chemotherapeutic agents. Brachydin A (BrA), a glycosylated flavonoid extracted from *Fridericia platyphylla*, displays a remarkable antitumoral effect against in vitro mPCa cells cultured as bidimensional (2D) monolayers. Considering that three-dimensional (3D) cell cultures provide a more accurate response to chemotherapeutic agents, this study investigated the antiproliferative/antimetastatic effects of BrA and the molecular mechanisms underlying its action in mPCa spheroids (DU145) in vitro. BrA at 60–100 μM was cytotoxic, altered spheroid morphology/volume, and suppressed cell migration and tumor invasiveness. High-content analysis revealed that BrA (60–100 µM) reduced mitochondrial membrane potential and increased apoptosis and necrosis markers, indicating that it triggered cell death mechanisms. Molecular analysis showed that (i) 24-h treatment with BrA (80–100 µM) increased the protein levels of DNA disruption markers (cleaved-PARP and p-γ-H2AX) as well as decreased the protein levels of anti/pro-apoptotic (BCL-2, BAD, and RIP3K) and cell survival markers (p-AKT1 and p-44/42 MAPK); (ii) 72-h treatment with BrA increased the protein levels of effector caspases (CASP3, CASP7, and CASP8) and inflammation markers (NF-kB and TNF-α). Altogether, our results suggest that PARP-mediated cell death (parthanatos) is a potential mechanism of action. In conclusion, BrA confirms its potential as a candidate drug for preclinical studies against mPCa.

## 1. Introduction

Prostate cancer (PCa) is the second most frequent cancer type in men and the fifth leading cause of death worldwide [1,2]. In 2020, the Global Cancer Observatory (Globocan) reported 1.414 million new cases of PCa, whose incidence depends on regions and population. After metastasis, PCa patients are usually treated with hormonal therapy by androgen deprivation therapy (ADT), namely abiraterone and enzalutamide, associated with chemotherapy [3]. The taxane agents docetaxel and cabazitaxel are the main chemotherapeutic drugs used to treat metastatic PCa (mPCa) due to their action on the cytoskeleton and induction of cell cycle arrest (G2/M)/cell death [4]. However, cellular events such as drug efflux, modifications in microtubule dynamics, autophagy, epithelial–mesenchymal transition, and the expression of microRNAs and anti-apoptotic proteins have increased mPCa chemotherapy resistance [5].

Plant-derived secondary metabolites (phytochemicals) have promoted therapeutic benefits for cancer treatment, with an important role in mPCa chemotherapy [6,7,8]. *Fridericia platyphylla* (Cham.) L.G. Lohmann (synonym *Arrabidaea brachypoda*) is a plant native to Brazil that produces unique dimeric glycosylated flavonoids [9]. Brachydins are dimeric flavonoids [10] with antiviral, anti-inflammatory, and antiparasitic properties [11]. The first flavonoid dimer isolated from *F. platyphylla*, Brachydin A (BrA; Figure 1), exerts significant cytotoxic/antiproliferative action in PC3 and DU145 mPCa cells in vitro, and selective effects to tumor cells, since it is less cytotoxic (7.3 times) to PNT2 non-tumor prostate cells [12,13]. Despite these promising results, there is still a need to analyze the antitumor effects of BrA in a more robust in vitro cell culture model to confirm its chemotherapeutic potential for mPCa.

Three-dimensional (3D) culture models are multicellular tumor spheroids with a structural and histomorphology organization, cell–cell/cell–matrix interactions, and authentic pathophysiological conditions similar to in vivo models [15,16]. Their features make tumor spheroids an excellent tool for testing drug efficacy and evaluating new antitumor compounds [17,18,19,20,21]. Specifically, tumor spheroids from the mPCa cell lines DU145, PC3, and LNCaP can provide a representative pathophysiological tumor heterogeneity model for therapeutic research [22]. Herein, we examined the antitumoral and antimetastatic potential of BrA in DU145 mPCa cells cultured as 3D tumor spheroids in vitro. Moreover, we analyzed the biological parameters of mitochondrial function, cell death, and oxidative stress using a high content screening (HCS) platform, as well as the expression of proteins related to cell proliferation, inflammation, and cell death to unravel the mechanisms of action of BrA.

## 2. Materials and Methods

### 2.1. Brachydin A (BrA)

The details regarding the collection, extraction, isolation, and identification of Brachydin A (BrA) were previously reported [10]. The Brazilian Genetic Heritage Management Council approved all procedures (Proc. nº A451DE4). Lyophilized BrA (purity ≥ 98%) was kindly provided by Prof. Claudia Q. da Rocha (Federal University of Maranhão; UFMA, Maranhão, Brazil). BrA was dissolved in dimethyl sulfoxide (DMSO; Sigma-Aldrich, St. Louis, MO, USA) to prepare a stock solution (MW: 524.00 g/L) that was kept in sterile tubes protected from light and stored at −20 °C. The BrA working solutions were prepared in DMSO to achieve the final concentration of 10, 20, 40, 60, 80, and 100 µM. All treatments were performed by replacing 50% of the culture supernatant with a 2× drug-supplemented culture medium.

### 2.2. Cell Lines and Culture Conditions

The prostate tumor DU145 cell line, isolated from a metastatic brain site, was purchased from ATCC^®^ (American Type Culture Collection—Cat. Nº HTB-81TM) and maintained in RPMI 1640 medium (Gibco—Thermo Fisher Scientific; Carlsbad, CA, USA) with 10% Fetal Bovine Serum (Gibco), 1% antibiotic/antimycotic mix (10,000 units/mL penicillin, 10,000 µg/mL streptomycin, and 25 µg/mL amphotericin B; Gibco) and 0.024% NaHCO_3_ (Sigma-Aldrich; St. Louis, MO, USA). DU145 cells were kept in a 3110 Series II CO_2_ water-jacketed incubator (Thermo Fisher Scientific; Carlsbad, CA, USA) at 37 °C with a 5% CO_2_ atmosphere and 96% relative humidity. All culture procedures followed the recommendations proposed by Bal-Price and Coecke [23]. All experiments were performed with aliquots between the 3rd and 8th cell passages. The cell line was authenticated by the Laboratory of Molecular Diagnosis of Barretos Cancer Hospital [24], using short tandem repeat (STR) DNA typing with an 8 STR loci panel (D5S818, D13S317, D7S820, D16S539, vWA, TH01, TPOX, and CSF1P0) plus gender determination (AMEL).

### 2.3. Tumor Spheroids Culture 

DU145 tumor spheroids were obtained using the agarose coated-overlay method, according to Friedrich et al. [25]. Briefly, 96-well plates (Greiner Bio-One; Monroe, NC, USA) were previously coated with 1.5% normal melting point agarose (NMP; Invitrogen —Thermo Fisher Scientific; Carlsbad, CA, USA) in incomplete RPMI 1640 medium. After solidification, DU145 cells (3.0 × 103) suspended in complete culture medium were added to each well using an 8-channel micropipette (HTL Discovery; Corning, Warszawa, Poland). The plates were transferred to the incubator and held immobile for 96 h (h) for tumor spheroids (300–500 µm^3^) formation (called initiation).

### 2.4. Cell Viability 

#### 2.4.1. Acid Phosphatase Assay

The acid phosphatase assay was performed according to Friedrich et al. [25] to quantify viable cells from the cytosolic acid phosphatase enzyme. After the initiation step (96 h), DU145 spheroids were treated with RPMI 1640 (negative control; NC), 1% DMSO (solvent control; SC), or BrA (10, 20, 40, 60, 80, and 100 µM) for 24, 48, and 72 h. Next, the spheroids were transferred to standard flat-bottom 96-well plates with an 8-channel manual pipette and centrifuged at 400× *g* for 10 min for adhesion to the well bottom. Then, the spheroids were washed carefully with PBS (pH 7.4) and suspended in acid phosphatase buffer (0.7 mL of 0.1 mol/L sodium acetate, 0.02 mL of triton X-100, 21.4 mL of distilled water, and 44 mg of immunopure p-nitrophenyl phosphate; Sigma-Aldrich). After incubating the plates for 90 min at 37 ºC, 1 mol/L NaOH was added to the wells, and the absorbance values were recorded in a microplate reader (Biotek Elx800—Winooski, VT, USA) at λ = 405 nm. The cell viability percentage (%) was obtained by comparing the absorbances of each treatment with the NC, which was considered as 100%. All analyses were performed with six spheroids/replicates (*n* = 6) and three biological experiments (*n* = 3).

#### 2.4.2. Resazurin Assay

The resazurin assay (AlamarBlue^®^; Sigma-Aldrich) was performed according to Riss et al. [2] and Walzl et al. [26] to assess cell viability from the resazurin dye reduction performed by dehydrogenase enzymes. After the initiation step, DU145 spheroids were treated with RPMI 1640 (NC), 1% DMSO (SC), or BrA (10–100 µM) for 24, 48, 72, and 168 h. Additional experiments were performed with Docetaxel (DTX; Sigma-Aldrich—1, 5, 10, and 50 µM) at 24, 48, and 72 h, as positive control (PC). Next, a resazurin working solution (0.15 mg/mL in PBS) was added to 20% of the final volume to each well containing the tumor spheroids. The plates were incubated for 24 h at 37 °C, and the absorbance was recorded in a microplate reader (Biotek Elx800) at λ = 570 nm for resofurin and λ = 600 nm for resazurin. The absorbance values were multiplied by the oxidation factor at each wavelength and subtracted from converted resofurin. The percentage of cell viability was calculated by normalizing the absorbance of each treatment with the NC, which was considered as 100%. All analyses were performed with six spheroids/replicates (*n* = 6) and three biological experiments (*n* = 3). 

### 2.5. Volume, Morphology, and Integrity Analyses 

The volume, morphology, and integrity of DU145 spheroids were analyzed based on the methods reported by Friedrich et al. [25] and Vinci et al. [27]. After the initiation step, photomicrographs of the DU145 spheroids were recorded using the Axio Cam MRc image capture system (Carl Zeiss; Göttingen, Germany) coupled to an inverted Axio LabA1 microscope (Carl Zeiss) using the 10× objective, and analyzed with the aid of the AxioVision SE64 Rel. 4.9.1 software (Carl Zeiss). Then, the spheroids were treated with RPMI 1640 (NC), 1% DMSO (SC), 50 µM DTX (PC), or BrA (10–100 µM). For integrity/morphology analysis, a second photomicrograph of each spheroid was obtained after 72 h of treatment. The other photomicrographs were obtained every 48 h until 168 h. After each photomicrograph, 50% of the culture medium in each well was replaced along with the treatments. In the integrity/morphology assessment, each image was analyzed to detect irregular spheroids (without circular shape), cell disaggregation, or irregular cell agglomeration. For cell volume quantification, the circumferences of tumor spheroids were analyzed with the AxioVision SE Rel. 4.9.1 software using the “measure” tool, and the area was reported in μm^3^. All analyses were performed with six spheroids/replicates (*n* = 6) in three biological experiments (*n* = 3).

### 2.6. Clonogenicity 

The clonogenicity assay with disaggregated cells from DU145 spheroids was performed according to Mikhail et al. [28]. The DU145 spheroids were initiated in 96-well plates and treated with RPMI 1640 (NC), 1% DMSO (SC), 50 µM DTX (PC), or BrA (40–100 µM) for 72 h. Then, six spheroids were transferred to a 1.5 mL centrifuge microtube to obtain the required number of cells, and centrifuged for 5 min at 300× *g*. After removing the supernatant, the cells were suspended in TrypLE™ Enzyme 1X (Gibco), incubated at 37 °C for 5 min, and manually shaken every 1 min. The cells were diluted with culture medium and centrifuged for 5 min at 300× *g*. The supernatant was discarded, the resulting pellet was suspended, and 250 cells were seeded in 6-well plates and incubated for seven days. The DU145 colonies were washed with PBS and fixed with methanol: acetic acid: water (1:1:8 mL; *v/v/v*; Sigma-Aldrich) for 30 min and stained for 25 min with 5% Giemsa diluted in phosphate buffer (1:20 mL; *v/v*). The cells were washed with distilled water, and the colonies were counted using a manual hematological counter. The plating efficiency and the survival fraction were calculated according to Franken et al. [29]. All analyses were performed with two replicates (*n* = 2) in three biological experiments (*n* = 3).

### 2.7. Cell Migration

The 3D cell migration analysis with DU145 spheroids was performed with ECM Gel^®^ (Cat #E1270; Sigma-Aldrich) based on the protocol proposed by Vinci et al. [30]. The commercial ECM Gel^®^ solution (10 mg/mL) was diluted in incomplete RPMI medium to the working concentration (200 μg/mL), pipetted into a 96-well plate, and incubated at room temperature for 3 h to fix the ECM to the well bottom. Subsequently, the remaining non-adhered ECM was carefully removed, and the wells were washed twice with PBS at room temperature and filled with 1% bovine serum albumin (BSA; Sigma-Aldrich) blocking solution diluted in PBS. After the initiation step, DU145 spheroids suspended in culture medium were transferred to the plates pre-coated with ECM gel for 30 min, and treated with RPMI 1640 (NC), 1% DMSO (SC), 50 µM DTX (PC), or BrA (40, 60, 80, and 100 µM) for a further 30 min. The DU145 spheroid images corresponding to time 0 (t0) at 24 h and 48 h were acquired. For analyzing cell migration, the circumference of the migrated cells around the spheroid was measured in each image using the AxioVision SE64 Rel. 4.9.1 software (Carl Zeiss) with the “measure” tool. The spheroid growth was analyzed by measuring its area comprised of the spheroid and cell migration presented in μm^3^. All analysis were performed with six spheroids/replicates (*n* = 6) in three biological experiments (*n* = 3).

### 2.8. Tumor Spheroid Invasion 

The 3D cell invasion assay was performed according to Vinci et al. [31] and Berens et al. [32]. Before carrying out the treatments, an adhesion medium containing bovine skin gelatin (BSG; Sigma-Aldrich) (3.75 mg/mL in culture) was prepared in a 15 mL tube and incubated in a water bath at 60 °C until solubilization. The DU145 spheroids were seeded in 96-well U-bottom plates (TPP; Trasadingen, Switzerland) containing 1.5% NMP agarose for 96 h. After the initial step, 50% culture medium was removed and the BSG adhesion medium was added to the microtubes, followed by treatments with BrA (40–100 µM), RPMI 1640 (NC), 1% DMSO (SC), or 50 µM DTX (PC). The adhesion medium (100 μL) containing the treatments were carefully added to the wells, the bubbles were removed, and the plates were transferred to a 5% CO_2_ incubator at 37 °C for 30 min for BSG solidification. An image for each tumor spheroid was recorded at 0, 24, 48, and 72 h intervals using the Zeiss Primo Vert microscope on the 4× objective and the Zen2 software (Zeiss). After capturing the images, the area covered by the spheroids was measured using the “measure” tool in μm^3^. The spheroid area (invaded area × spheroid area) at each time point was calculated concerning the area at t = 0, and the data were converted into percentage (% t0). All analyses were performed with four spheroids/replicates (*n* = 4) in three biological experiments (*n* = 3). 

### 2.9. Cell Viability/Death Rate (Flow Cytometry)

Flow cytometry analysis was performed using the Dead Cell Apoptosis Kit with annexin V-FITC and propidium iodide (PI) (Cat Nº V13242; Invitrogen) after 72 h of treatment with BrA (40–100 µM), RPMI 1640 (NC), 1% DMSO (SC) or 50 µM DTX (PC). Then, the culture medium (80 μL) was removed, and the remaining medium with the spheroids was transferred to a 2.0 mL microtube. One sample was composed of eight spheroids to obtain a sufficient cell number. Thus, the spheroids were centrifuged for 5 min at 300× *g*, the supernatant discarded, and TrypLE enzyme (1X) was added to each tube. The tubes were incubated in a 5% CO_2_ incubator at 37 °C for 10 min and manually shaken every 2 min. The cells were diluted with complete culture medium and centrifuged for 5 min at 300× *g*. The cell pellet was suspended in annexin-binding buffer (1X), transferred to 5 mL flow cytometry tubes (Falcon^®^ Round-Bottom Polystyrene Tubes; Corning, Tewksbury, MA, USA), and incubated with annexin V-FITC for 15 min in an ice bath. Next, PI (2 μg/mL) was added to the tubes and the samples were immediately analyzed in the BD LSR Fortessa^®^ Flow Cytometer (BD Biosciences, San Jose, CA, USA) under an excitation λ = 488 nm and emission λ = 530/575 nm laser (or equivalent). Viable cells were not stained with annexin-V or PI; early apoptotic cells were labeled only with annexin-V; late apoptotic cells were labeled with both annexin-V and PI; necrotic cells were marked only with PI. All analyses were performed with 10,000 events from eight spheroids/replicates (*n* = 8) in three biological experiments (*n* = 3).

### 2.10. High-Content Screening (HCS)

#### 2.10.1. Apoptosis

All apoptosis analyses were performed using the HCS platform In-Cell Analyzer 2200 (GE Healthcare Life Sciences, Chicago, IL, USA) with the CellEvent Caspase 3/7 Green reagent (Cat. Nº. C10423; Invitrogen), following the manufacturer’s instructions. The DU145 spheroids were started in 96-well black plates with U-shaped transparent bottoms (CellStar^®^ Chimney Well µClear^®^; Greiner Bio-One; Monroe, NC, USA) filled with complete medium and maintained for 96 h in a cell culture incubator. After initiation, the spheroids were treated with RPMI 1640 (NC), 1% DMSO (SC), 50 µM DTX (PC), or BrA (40–100 µM). Along with the treatments, CellEvent Caspase 3/7 reagent (2.5 μM in Hank’s solution) and Hoechst 33342 (10 μM in PBS) were added to all the wells. The samples were monitored for 24, 48, and 72 h using the HCS platform. The acquisition parameters were “brightfield” (Channel 1), FITC (Channel 2; CellEvent Caspase 3/7), and DAPI (Channel 3; Hoechst 33342). The “autofocus” was set to “brightfield”, and the plates were scanned using the acquisition mode “one field of view per well” with the 4× objective. The acquired images were analyzed using the In-Cell Investigator v.1.3 software (GE Healthcare; Chicago, IL, USA). Apoptotic cells were quantified using the “average fluorescence intensity” recorded in the FITC-channel. Representative images from DU145 spheroids were acquired by merging Hoechst (Blue/DAPI) and Caspase 3/7 (Green/FITC). All analyses were performed with four spheroids/replicates (*n* = 4) in three biological experiments (*n* = 3).

#### 2.10.2. Necrosis 

Necrotic cells were analyzed using the HCS platform In-Cell Analyzer 2200 (GE Healthcare) with PI (Thermo Fischer Scientific) dye, following the manufacturer’s instructions. The experimental design and treatments were performed as described for the apoptosis assay reported above. Along with the treatments, PI (10 μg/mL in PBS) and Hoechst 33342 (10 μM in PBS) were added to each well, and the samples were analyzed at 24, 48, and 72 h of treatment. The acquisition parameters were “brightfield” (Channel 1), Cy3 (Channel 2; PI), and DAPI (Channel 3; Hoechst 33342). The “autofocus” was set to “brightfield,” and the plates were scanned with acquisition mode “one field of view per well” in the 4× objective. The acquired images were analyzed using the In-Cell Investigator v1.3 software (GE Healthcare). Necrotic cells were quantified using the “average fluorescence intensity” recorded on the Cy3 channel. Representative images from DU145 spheroids were acquired by merging Hoechst (Blue/DAPI) and PI (Orange/Cy3). All analyses were performed with four spheroids/replicates (*n* = 4) in three biological experiments (*n* = 3).

#### 2.10.3. Mitochondrial Membrane Potential 

All images for the mitochondrial membrane potential (Δψm) analysis were acquired using the HCS platform In-Cell Analyzer 2200 (GE Healthcare) with the MitoStatus Red™ dye (Cat. Nº. 564697; BD Pharmingen™), following the manufacturer’s recommendations. The experimental design and treatments were performed as described for the apoptosis assay reported above. Along with treatments, MitoStatus Red (200 nM in DMSO) and Hoechst 33342 (10 μM in PBS) were added to each well and analyzed at 0.5, 4, 24, 48, and 72 h. The acquisition parameters were “brightfield” (Channel 1), Cy5 (Channel 2; MitoStatus), and DAPI (Channel 3; Hoechst 33342). The “autofocus” was set to “brightfield,” and the plates were scanned with the acquisition mode “one field of view per well” with 4× objective. The acquired images were analyzed using the In-Cell Investigator v1.3 software (GE Healthcare). The Δψm potential was quantified using the “average fluorescence intensity” recorded in the Cy5 channel. Representative images of DU145 spheroids were acquired by merging Hoechst (Blue/DAPI) and MitoStatus (Pink/Cy3). All analyses were performed with four spheroids/replicates (*n* = 4) and three biological experiments (*n* = 3).

#### 2.10.4. Mitochondrial Redox Status

All images for mitochondrial redox status analysis were acquired on the HCS platform In-Cell Analyzer 2200 (GE Healthcare) using the MitoSOX™ Red Mitochondrial Superoxide Indicator reagent (Cat. Nº. M36008; Invitrogen), following the manufacturer’s instructions. The experimental design and treatments were performed as described for the apoptosis assay reported above. Along with the treatments, MitoSOX™ Red (5 μM in DMSO) and Hoechst 33342 (10 μM in PBS) were added to each well and analyzed at 24, 48, and 72 h. The acquisition parameters were “brightfield” (Channel 1), TexasRed (Channel 2; MitoSox), and DAPI (Channel 3; Hoechst 33342). The “autofocus” was performed in “brightfield,” and the scanned plates with acquisition mode “one field of view per well” with the 4× objective. The images were analyzed using the In-Cell Investigator v1.3 software (GE Healthcare). Mitochondrial superoxide activity was determined based on the “average fluorescence intensity” recorded on the TexasRed channel. Representative DU145 spheroids images were acquired by merging Hoechst (Blue/DAPI) and MitoSox (Red/TexasRed). All analyses were performed with four spheroids/replicates (*n* = 4) in three biological experiments (*n* = 3).

### 2.11. Protein Expression

Protein expression was analyzed by western blotting after 24 and 72 h of treatment with BrA (80–100 µM) or 1% DMSO (SC). After 24 h of exposure, whole-cell lysates from DU145 spheroids were obtained with Lysis Buffer (50 mM Tris pH 7.6–8, 150 mM NaCl, 5 mM EDTA, 1 mM Na3VO4, 10 mM NaF, 10 mM Na4P2O7, 1% NP-40, Sigma-Aldrich) and Protease Inhibitors Cocktail (Roche, Cat. Nº 11697498001). Next, proteins (20 µg) were separated by standard 10–15% SDS-PAGE gel electrophoresis, transferred to a nitrocellulose membrane, and incubated overnight with primary antibodies against BCL-2, BAD, RIP3K, PARP, p-γ-H2AX, p-AKT, p-ERK1/2 (dilution 1:1000 µL), or α-tubulin (dilution 1:2000 µL), which was used as the loading control. All antibodies were acquired from Cell Signaling Technology (Danvers, MA, USA). The membranes were incubated with horseradish peroxidase-conjugated secondary antibodies (1:5000 µL) and revealed using ECL Western Blotting Detection Reagents (GE Healthcare). The enhanced chemiluminescence was detected using the ImageQuant LAS 4000 mini documentation system (GE Healthcare). After 72 h of exposure, whole-cell lysates from DU145 spheroids were obtained with RIPA Lysis and extraction buffer containing Halt Protease Inhibitor Cocktail (Thermo Fisher Scientific). Then, protein samples (10 µg) were treated with Bolt 4X LDS Sample Buffer and 10X Bolt™ Sample Reducing Agent (Thermo Fisher Scientific) and denatured at 70 °C for 10 min. The proteins were separated using the NuPAGE™ 4–12% Bis-Tris Gel and MiniGel Tank system (Thermo Fisher Scientific) and further transferred to PVDF and nitrocellulose (iBlot™ Transfer Stacks) membranes using the iBlot2 Gel Transfer Device (Thermo Fisher Scientific). Immunodetection was performed using primary antibodies (1:500 µL) against NF-kB/p65, CASP8, CASP7, CASP3, BAX, TNF-α (Abclonal, Woburn, MA, USA), or β-Actin (Abcam, Waltham, MA, USA), which was used as the loading control. After this, the membranes were incubated with goat anti-rabbit IgG H&L conjugated with horseradish peroxidase (1:5000 µL; Abclonal) and revealed using the SuperSignal™ West Pico Substrate (Thermo Fisher Scientific). The chemiluminescence signals were detected using the L-Pix Chemi (Loccus Biotechnology, Cotia, SP, Brazil, Cat. Nº 03608602) system coupled with L-Pix Image software (Loccus Biotechnology). The densitometric values were calculated with the aid of the ImageJ software (NIH.gov accessed on 6 March 2021). All analysis were performed with ninety-six spheroids/replicates (*n* = 96) in two biological experiments (*n* = 2).

### 2.12. Statistical Analysis

The results were initially submitted to data distribution analysis (normality test) using the Shapiro–Wilk test. All samples showed parametric distribution, and the data were analyzed using analysis of variance (ANOVA) followed by Dunnett’s post-test, considering *p* ≤ 0.05 as a minimal significant value. All statistical analyses were performed using the GraphPad Prism 7.0 software (La Jolla, CA, USA). 

## 3. Results

### 3.1. BrA Is Cytotoxic to DU145 Prostate Tumor Spheroids

The cytotoxic effects of BrA flavonoid in DU145 spheroids were evaluated using the acid phosphatase and resazurin assays. In the former assay, BrA decreased cell viability at exposure times equal to or greater than 48 h and at concentrations ≥ 60 µM (Table 1). Next, we selected the chemotherapeutic agent DTX at 50 µM (Appendix A) as the positive control for the following experiments. In the resazurin assay, BrA was cytotoxic only at 100 μM after 24 h exposure (Table 2). However, at 60 μM, BrA was sufficient to decrease cell viability after 48 and 72 h of treatment, while it induced cytotoxicity after 168 h of exposure at 40 μM (Table 2). Collectively, BrA at ≥40 μM was cytotoxic to DU145 tumor spheroids in both tests at 48 h of treatment; hence, this was the lowest concentration selected for the subsequent experiments.

### 3.2. BrA Alters Morphology/Volume of DU145 Spheroids

After evaluating the cytotoxic effects of BrA, we analyzed whether it altered the volume, morphology, and integrity of DU145 spheroids (Figure 2). BrA at 40, 60, 80, and 100 µM increased spheroid growth rate up to 72 h, but it promoted cell breakdown in the central spheroid region after 120 h exposure (Figure 2A,B). Accordingly, BrA (40–100 µM) reduced the survival fraction of DU145 cells disaggregated from spheroids after 72 h exposure (Figure 2D). These results suggest the effectiveness of BrA in antitumoral therapy against quiescent cells that possibly resisted the treatments in DU145 spheroids.

### 3.3. BrA Inhibits Cell Migration and Tumor Invasiveness

Besides its antiproliferative potential, BrA flavonoid may prevent tumor cells to generate metastases that are commonly resistant to chemotherapy. The cell migration (Figure 3) and tumor invasiveness (Figure 4) assays were performed with DU145 spheroids to examine whether BrA prevented the metastatic process. In the cell migration assay (Figure 3), BrA at 60–100 µM decreased the migration area after 24 h of treatment, while higher BrA concentrations (>80 µM) almost completely suppressed cell migration to ECM gel. BrA inhibited tumor invasiveness after 48 h (Figure 4), mainly at concentrations > 60 µM (Figure 4B). Collectively, our results demonstrated that BrA at ≥80 µM suppressed both cell migration and tumor invasiveness—two important metastatic mechanisms—in mPCa spheroids.

### 3.4. BrA Induces Apoptosis Cell Death 

After demonstrating the cytotoxicity, phenotype alterations, and antimetastatic properties of BrA flavonoid in DU145 spheroids, we examined how BrA induced cell death using annexin-V and PI by flow cytometry (Figure 5). After 72 h, BrA at concentrations > 60 µM induced early (annexin-V) and late apoptosis (annexin-V + PI) (Figure 5A,B), confirming the decreased viability reported above (Table 1 and Table 2 and Figure 1). Remarkably, more than 61% of cells were apoptotic (early + late) after treatment with higher BrA concentrations (>80 µM).

### 3.5. HCS Demonstrates Mitochondrial Depolarization and Apoptotic/Necrotic Cell Death 

We performed HCS analysis in DU145 spheroids (Figure 6) to further characterize the BrA mechanism of action. Initially, BrA did not induce detectable alterations in the MitoSOX Red fluorescence in DU145 spheroids (Figure 6A,B), but it altered their Δψm potential after 0.5 h of treatment (Figure 6C,D). Accordingly, the Caspase 3/7 marker (Figure 6E,F) indicated that BrA > 60 µM increased the apoptosis rate in DU145 spheroids after 24 and 48 h of treatment. Higher BrA concentrations (>80 µM) elevated the PI fluorescence intensity after 24–72 h of treatment (Figure 6G,H), indicating necrotic cell death. Together, our results suggest that BrA decreases viability by inducing mitochondrial depolarization, apoptosis, and necrosis.

### 3.6. BrA Initially Up-Regulates DNA Disruption Proteins Followed by Apoptosis-Related Proteins 

Considering that BrA flavonoid induces mitochondrial membrane depolarization, apoptosis, and necrosis, which affected cell viability, cell migration, and tumor invasion in DU145 spheroids, we examined which molecular mechanisms underlie these effects. We selected the higher BrA concentrations (80 and 100 µM) to analyze several markers of cell death (apoptosis, necrosis, and necroptosis) and cell proliferation pathways (Figure 7). After 24 h exposure, BrA upregulated the expression of DNA disruption proteins (cleaved-PARP1 and p-γ-H2AX) and downregulated the expression of proteins related to cell death (BCL-2, BAD, and RIP3K) and cell proliferation (p-AKT and p-ERK1/2) (Figure 7A,B). At 72 h, BrA increased apoptosis and inflammation proteins (CASP3, CASP7, CASP8, NF-kB/p65, and TNF-α) (Figure 7C,D). Altogether, our findings suggest that BrA is a promising agent for cancer therapy because its mechanisms of action on mPCa cells resemble those of widely used chemotherapeutic drugs.

## 4. Discussion

Metastatic prostate cancer (mPCa) has been treated with low molecular weight chemotherapeutics drugs that selectively induce cell death [33]. Taxanes are the most used chemotherapy drugs for castration-resistant mPCa and stabilize microtubules, preventing cellular division and resulting in cell-cycle arrest/cell death [4,34]. However, many biological mechanisms make mPCa cells resistant to taxanes [5], which prompts the continuous search for new agents. Natural compounds are sought due to their low toxicity to healthy cells and high performance to induce cell death pathways and inhibit tumor growth [35]. In this sense, phytochemicals have been presented as an alternative in the search for new molecules with unknown therapeutic potential.

In Brazil, the native plant *F. platyphylla* (Cham.) L.G. Lohmann (synonym *Arrabidaea brachypoda*), popularly known as “cipó-una”, produces unique dimeric glycosylated flavonoids named brachydins, with antimicrobial, anti-inflammatory, and antiparasitic activities [9,10,36]. BrA, the first dimeric flavonoid isolated from this plant species, exerts antitumoral activity by promoting cytotoxicity and cell death via apoptosis/necrosis in metastatic prostate PC3 cells in vitro [12]. Similarly, BrA decreases cell migration and induces apoptosis in prostate tumor DU145 cells, and is seven times more cytotoxic to tumor cells than to non-tumor prostate PNT2 cells (IC50 = 77.7 μM and 10.7 μM for DU145 and PNT2 cells, respectively) [13]. Therefore, the antitumoral action of the flavonoid BrA is selective to mPCa cells, with no relevant effects on non-tumor prostate cells.

Nevertheless, mPCa cells cultured in 2D monolayers have no structural/functional properties and may not reproduce in vivo functions (i.e., tumor microenvironment) [37]. On the other hand, tumor spheroids reproduce the tumor microenvironment and share morphological, functional, and transport properties with in vivo tissue [38,39]. In the present study, BrA was cytotoxic to DU145 tumor spheroids at ≥60 µM after 48 h exposure. These values were tenfold higher than those reported for PC3 and DU145 cells (6 µM) cultured at 2D monolayers [12,13]. Our data corroborated the estimated values for 3D cultures, which can be 5–50 times higher than those obtained in 2D culture models [40].

BrA flavonoid affected the volume/morphology of DU145 spheroids after 72 h exposure. The analysis of tumor spheroids is counted among the most relevant tools for evaluating the effectiveness of chemotherapy drug candidates [20]. Besides, BrA also decreased the survival fraction of DU145 cells disaggregated from the tumor spheroids after 72 h of treatment. The clonogenicity assay approach in tumor cells disaggregated from the spheroids is even more critical, as it can verify the existence of possible quiescent/senescent cells in the spheroid that resisted the treatment and started to proliferate again. This tumor quiescence/senescence state is known to arise after chemotherapy and is one of the underlying mechanisms of antitumor failure/therapy resistance [41,42].

Currently, many candidate molecules have antiproliferative potential, but tumor cells are able to escape the primary tumor and generate metastases that are commonly resistant to chemotherapy. Therefore, the search for new chemotherapeutic agents is based on finding molecules that can delay metastases formation [43]. The cell migration assay reported here demonstrated that BrA (40–100 µM) decreased the migratory area from DU145 spheroids after 24 h exposure. Beyond cell migration, tumor cells can also invade surrounding tissues during malignant progression and penetrate the basement membrane through the “invadopodia”, structures responsible for ECM degradation and vascular/lymphatic dissemination [44,45,46]. In the tumor invasion assay reported herein, treatment with BrA (40–100 µM) for 48 h decreased the invasive potential of DU145 spheroids. The results corroborate previous data showing that BrA (6 μM) decreases cell migration of DU145 cells at monolayers after 48 h exposure in the wound healing assay [12,13].

Besides migration and cell invasion, another biological factor that affects chemoresistance in mPCa is the cell death resistance promoted by the high expression of anti-apoptotic proteins [35]. Nowadays, the primary therapeutic strategies aim at new molecules that induce cell death in cancer cells [47]. BrA at ≥60 µM promoted apoptosis in DU145 spheroids after 72 h exposure. However, as flow cytometry analysis requires single cells, the tumor spheroids must be dissociated. The difficulty associated with dissociation due to strong cell–cell interactions is that it requires high enzyme concentrations (i.e., trypsin) or long periods of time (for cell separation), which can affect cell viability [48]. Alternatively, HCS platforms can provide multiparametric analysis without the dissociation of tumor spheroids [49], and it is frequently used to seek new antitumor drugs in 3D models [50,51,52].

The HCS analysis performed with MitoStatus Red™, a cationic fluorescent dye that accumulates in the mitochondria of viable cells but not in depolarized mitochondria, revealed that BrA (40–100 µM) decreased the Δψm of DU145 cells in tumor spheroids after 30 min (0.5 h) of treatment. The depolarization of Δψm is one of the first events in the apoptotic cascade; after triggering it, cytochrome c (Cyt C) is released into the cytoplasm, with the formation of Apoptotic peptidase activating factor 1 (Apaf-1), and subsequent caspase-9 activation [53]. The oxidative stress in mitochondria can lead to Δψm depolarization, and as a result, the superoxide anion (O_2_^•−^) is predominantly produced during oxidative phosphorylation [54]. Nevertheless, the HCS analysis using MitoSOX Red (a fluorogenic dye that O_2_^•−^ can oxidize in living cell mitochondria [55]) demonstrated that BrA did not have pro-oxidative effects on tumor prostate spheroids after 24, 48, and 72 h exposure. Previously, it was shown that BrA does not induce reactive oxygen species (ROS) overproduction in PC3 cells cultured in 2D monolayers [12].

In the apoptosis cascade, after Δψm depolarization, Cyt c release, and Apaf-1 formation, the activation of “apical” caspases 2, 8, 9, and 10 directly cleaves “effector” caspases; caspases-3 and 7 are the main proteolytic enzymes of intracellular substrates in the final steps of cell death [56]. The CellEvent Caspase-3/7 Green reagent can estimate the activation of caspase-3/7 in apoptotic cells through a fluorogenic signal [57]. Furthermore, PI is a red-fluorescent DNA intercalating dye to which the membrane of viable cells is impermeable and may indicate when necrosis death arises from severe homeostasis dysregulation or late apoptosis, where proteases have already degraded cell compartments and the nuclear membrane was disrupted, releasing its content into the extracellular compartment [58]. In DU145 spheroids, BrA increased the fluorescence from CellEvent caspase 3/7 after 24 h of treatment and PI labeling after 48 h of treatment. HCS results demonstrated that BrA (≥60 µM) exerted its antitumor effect in DU145 spheroids initially by mitochondrial Δψm depolarization, followed by cell death induction with apoptosis/necrosis phenotype. BrA (1.5, 6, and 24 μM) induces cell death by apoptosis/necrosis in PC3 and DU145 cells at 2D monolayers, as assessed using the triple staining assay with PI, Hoescht 33342, and Fluorescein [12,13]. Interestingly, BrA (1–60 µM) does not induce cell death (apoptosis or necrosis) in non-tumor epithelial prostate (PNT2) cells after 24 h exposure [13], indicating that it specifically promotes the cell death of mPCa cells.

Besides phenotypic results obtained with the HCS platform in DU145 spheroids, molecular analysis is essential to describe mechanisms of action. BrA downregulates the expression of phospho-AKT and up-regulates cleaved-PARP protein expression in metastatic prostate PC3 cells in 2D monolayers [12]. However, tumor spheroids have different gene expression profiles than cells grown in 2D models under the same conditions. Prostate spheroids express genes related to survival, inflammation, angiogenesis, migration, and differentiation which are overexpressed when compared with cell monolayers [17]. In this study, we examined the protein expression profile (western blotting technique) of cell death (apoptosis, necrosis) and cell proliferation genes in DU145 spheroids after a short-time treatment (24 h) with BrA (80–100 µM). The flavonoid BrA upregulated DNA disruption proteins (cleaved-PARP and phospho-γ-H2AX) and downregulated anti/pro-apoptotic (BCL-2, BAD, and RIP3K) and cell survival (phospho-AKT1 and phospho-44/42 MAPK) proteins.

Altogether, the results from imaging (HCS) and molecular (western blotting) assays suggest that the parthanatos cell death pathway was the possible mechanism of action with respect to BrA in DU145 spheroids. Certainly, cleaved-PARP is a point that directs death receptor signaling for either apoptosis, necrosis, necroptosis, or parthanatos [59,60]. Furthermore, another potential mechanism to cleaved-PARP accumulation is the response to genotoxic damage [61,62]. To investigate this hypothesis, DU145 spheroids were treated with BrA for 4 h (harsh exposure) and analyzed using the comet assay, a technique used for detecting alkylated bases and DNA single/double-strand breaks (SSB/DSBs) [63]. We found that BrA (40–100 µM) did not promote significative comet formation, indicating that increases in cleaved-PARP expression are related to cell death induction (Appendix A) and not with a possible genotoxic effect of BrA that could be leading to cell death. Previous analysis performed by our research group in 2D cultures with PC3 cells reported no genotoxicity with respect to BrA, as assessed using the comet assay, and cleaved-PARP overexpression mediated by cell death induction [12].

Parthanatos cell death is associated with (i) initial mitochondrial Δψm permeabilization (after 30 min); (ii) large-scale DNA fragmentation by cleaved-PARP and phospho-γ-H2AX overactivation (in apoptosis, it is typically observed at a small-to-moderate scale) [64]; and (iii) the downregulation of survival (MAPK/AKT1), anti-apoptotic (BCL-2), pro-apoptotic (BAD), and necroptosis (RIP3K) proteins. Specifically, BAD and RIPK3 proteins are critical in apoptosis and necroptosis-programmed cell death [65]. Therefore, it appears that there is excessive activation of cleaved-PARP due to the possible release of NAD+ (nicotinamide adenine dinucleotide) or AIF (apoptosis-inducing factor) from the mitochondrial depolarization. The AIF released from mitochondria can directly interact with free PARP or MIF (macrophage migration inhibitory factor) in the cytoplasm, forming the AIF/MIF complex, which follows DNA fragmentation and chromatin condensation (observed by phospho-γ-H2AX), leading to parthanatos [59].

Likewise, this correlation of up-regulated DNA disruption proteins (cleaved-PARP and phospho-γ-H2AX) after 24 h exposure and apoptosis/necrosis mixed-effects after 72 h exposure may lead to results that indicate a close connection between parthanatos and other cell death pathways, i.e., the co-targeting of these distinct death pathways, together with cleaved-PARP up-regulation [59]. In our study, BrA induced both apoptosis (CASP3/CASP7 marker) and necrosis (PI staining) in DU145 spheroids after 48 and 72 h of treatment and “late” apoptosis after 72 h of treatment. To better understand these data, we quantified apoptosis (CASP3, CASP7, and CASP8) and inflammation (NF-kB and TNF-α) proteins after 72 h treatment. BrA upregulated the expression of all proteins. In other words, the metabolic stress (i.e., NAD+ consumption) in response to parthanatos for damaged cell survival [66] in DU145 spheroids activates inflammation (TNF-α and NF-kB) proteins and subsequently initial caspases (5, 6, 7, 8) [58]. Thus, CASP8 activated the CASP3/7 cleavage, which promotes phosphatidylserine exposure, nuclear condensation, and membrane blebbing [53]. Our results from cell migration/tumor invasion inhibition, mainly after 48 and 72 h of treatment, also reinforce the occurrence of parthanatos since this mechanism downregulates epithelial–mesenchymal transition factors associated with tumor metastasis [59]. Below, we propose an action model for BrA flavonoid (Figure 8).

It is essential to highlight that the findings from the present study were obtained with DU145 cells, which can form well-differentiated tumor spheroids in vitro that can point to an even more invasive potential [18,67]. However, DU145 cells have the PTEN tumor suppressor gene actively expressed, and mutations of the PTEN gene lead to increased cell survival and chemoresistance in mPCa tumors, with worse patient prognoses [68]. The results obtained with BrA flavonoid in DU145 spheroids are promising, but we still need to evaluate this phytochemical in PTEN mutated mPCa cells (i.e., PC3 cells).

## 5. Conclusions

Here we demonstrated that the phytochemical BrA has cytotoxic, antiproliferative, anti-migratory, and anti-invasive effects that are possibly mediated by PARP overactivation (parthanatos) cell death in DU145 tumor spheroids in vitro. Thus, BrA has promising potential for preclinical studies of mPCa treatment.

## Figures and Tables

**Figure 1 pharmaceutics-14-00963-f001:**
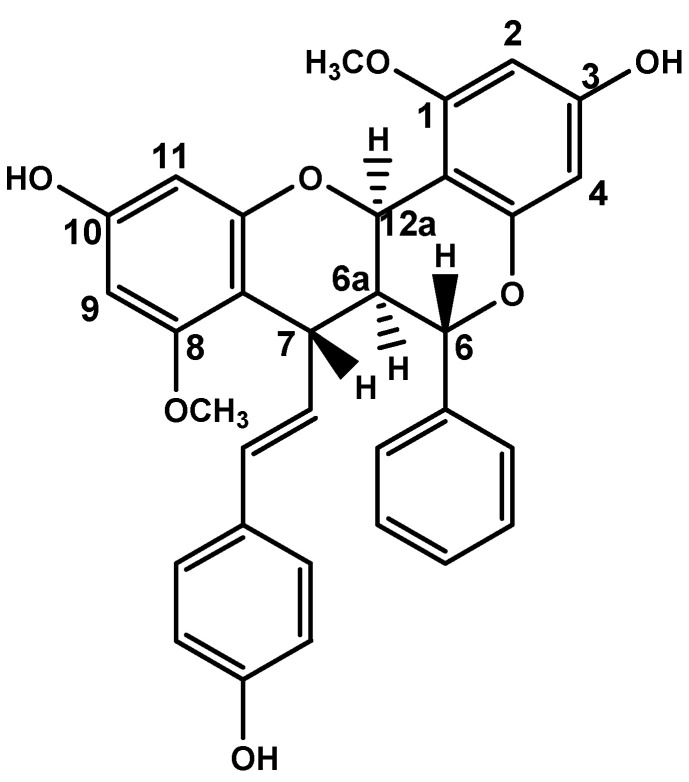
Chemical structure from the Brachydin A (BrA) flavonoid. The image was adapted from PubChem (CID 102339051) [14].

**Figure 2 pharmaceutics-14-00963-f002:**
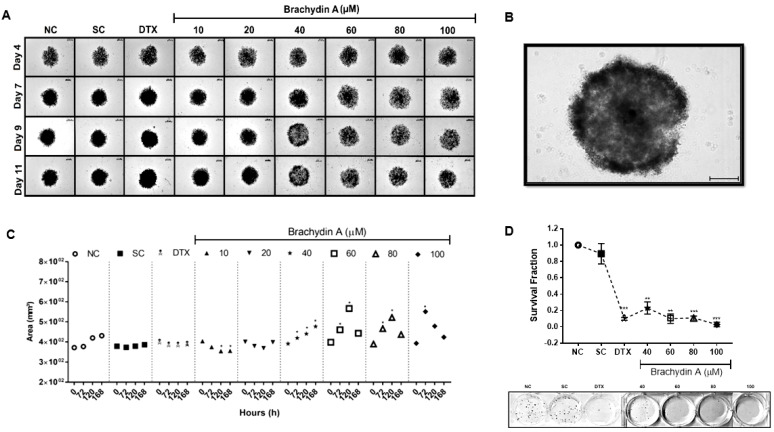
(**A**,**B**) Morphology and integrity of DU145 spheroids after treatment with Brachydin A (BrA) for 0 (Day 4), 72 (Day 7), 120 (Day 9), and 168 h (Day 11). All images were obtained using the Axio Cam MRc capture system coupled to Axio LabA1 microscope, using the 10× (**A**) and 40× (**C**) objectives. (**C**) Volume (area/µm^3^) of DU145 spheroids after 0 (Day 4), 72 (Day 7), 120 (Day 9), and 168 h (Day 11) of exposure with BrA and respective controls. (**D**) Survival fraction of DU145 cells disaggregated from spheroids treated for 72 h with BrA and their respective controls. All data points from volume/morphology represent the mean of six (*n* = 6) spheroids/replicates analyzed in three biological experiments (*n* = 3). * Values statistically different from the NC group at the point (date) (* *p* < 0.05; ** *p* < 0.01; *** *p* < 0.001; ANOVA followed by Dunnett’s post-test). NC: negative control (RPMI 1640); SC: solvent control (1% DMSO); DTX: Docetaxel (50 μM; positive control). Scale: 200 μm.

**Figure 3 pharmaceutics-14-00963-f003:**
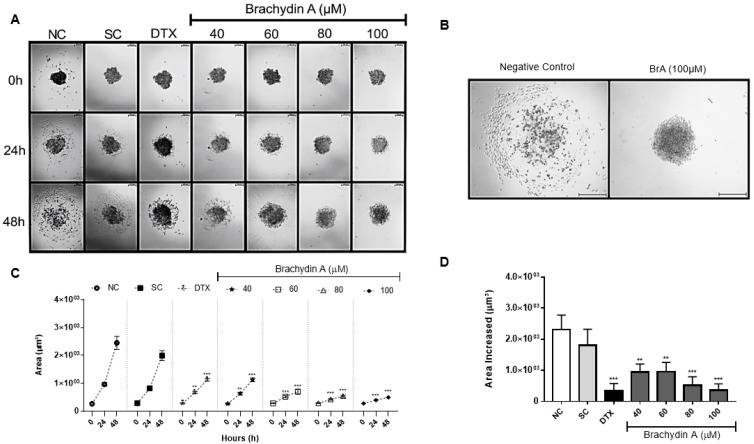
(**A**,**B**) Photomicrographs from DU145 spheroids in ECM gel obtained after 0, 24, and 48 h of exposure with Brachydin A (BrA) and respective controls. All images were acquired using the Axio Cam MRc capture system coupled to Axio LabA1 microscope, using the 10× objective, and analyzed using the AxioVision 3.1 software. Scale: 200 μm. (**C**,**D**) Cell migration area (µm^3^) from DU-145 spheroids after 0, 24, and 48 h exposures with BrA and their respective controls. All points/bars represent the mean ± standard deviation of the covered area from six (*n* = 6) spheroids/replicates in three biological experiments (*n* = 3). * Values statistically different from the NC group at the time-point (date) (** *p* < 0.01; *** *p* < 0.001; ANOVA followed by Dunnett’s post-test). NC: negative control (RPMI 1640); SC: solvent control (1% DMSO); DTX: Docetaxel (50 μM; positive control).

**Figure 4 pharmaceutics-14-00963-f004:**
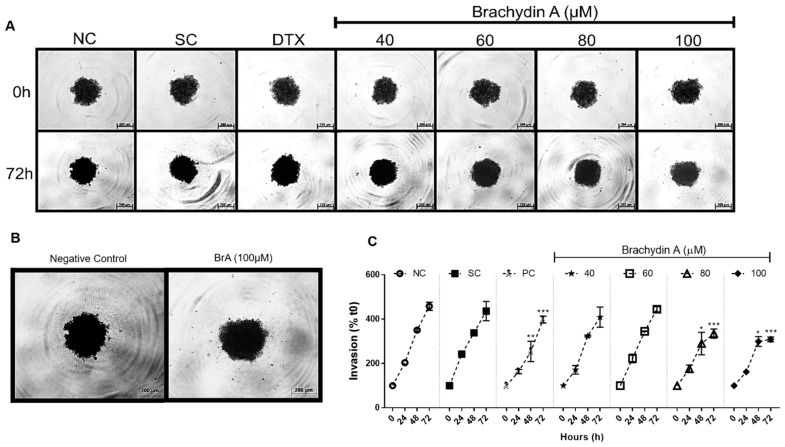
(**A**,**B**) Photomicrographs of DU145 spheroids in bovine skin gelatin after 0 and 72 h of exposure with Brachydin A (BrA) and respective controls. The images were acquired using the Axio Cam MRc image capture system coupled to an Axio LabA1 microscope, using the 10× objective, and analyzed using the AxioVision 3.1 software. Scale: 200 μm. (**C**) Percentage (%) of cell invasion (invadopodia) formed by DU145 spheroids after 24, 48, and 72 h of exposure with BrA and respective controls. The invasion area (µm^3^) into the ECM (gelatin) was converted to % considering the area increase at 0 h. All data points represent the mean ± standard deviation of four (*n* = 4) spheroids/replicates in three biological experiments (*n* = 3). * Values statistically different from the NC group at that point (date) (* *p* < 0.05; ** *p* < 0.01; *** *p* < 0.001; ANOVA followed by Dunnett’s post-test). NC: negative control (RPMI 1640); SC: solvent control (1% DMSO); DTX: Docetaxel (50 μM; positive control). Scale: 200 μm.

**Figure 5 pharmaceutics-14-00963-f005:**
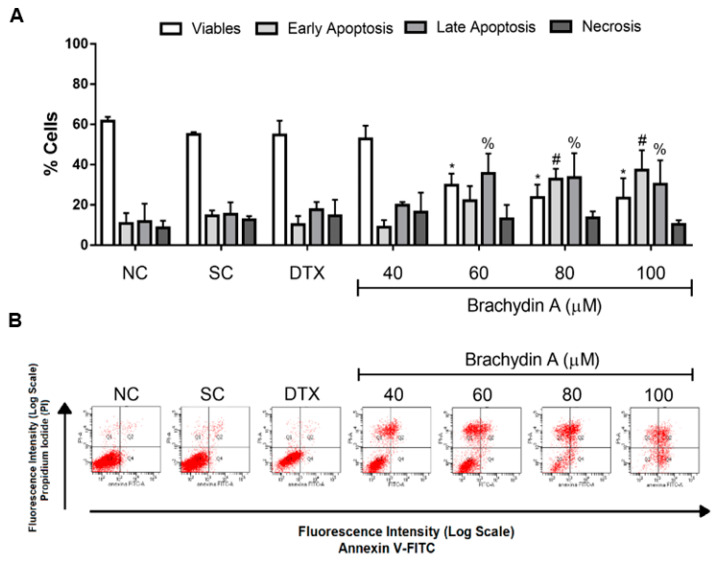
Flow cytometric analysis of DU145 spheroids treated with Brachydin A (BrA) for 72 h and respective controls and stained with Annexin V-FITC/propidium iodide (PI). (**A**) Percentage (%) of viable, early apoptotic, late apoptotic, and necrotic cells in DU145 spheroids after 72 h exposure with Brachydin A (BrA). The bars represent the mean ± standard deviation of eight (*n* = 8) spheroids/replicates disaggregated in three biological experiments (*n* = 3). *, ^#^ Values statistically different from the NC group (*p* < 0.05; two-way ANOVA followed by Dunnett’s post-test). (**B**) Representative dot-plots of DU145 spheroids treated for 72 h with BrA. Lower left quadrant (Q3): negative cells for both Annexin V-FITC and PI; lower right quadrant (Q4): cells labeled with Annexin V (early apoptotic cells); upper left quadrant (Q1): cells labeled only with PI (necrotic cells); right upper quadrant (Q2): cells labeled with Annexin V and PI (late apoptotic cells). The bars represent the mean ± standard deviation of experiments with six (*n* = 6) spheroids/replicates and three biological experiments (*n* = 3). Values statistically different from the NC group for * early apoptosis; ^#^ late apoptosis and ^%^ necrosis (*p* < 0.05; ANOVA followed by Dunnett’s post-test). NC: negative control (RPMI 1640); SC: solvent control (1% DMSO); DTX: Docetaxel (50 μM; positive control).

**Figure 6 pharmaceutics-14-00963-f006:**
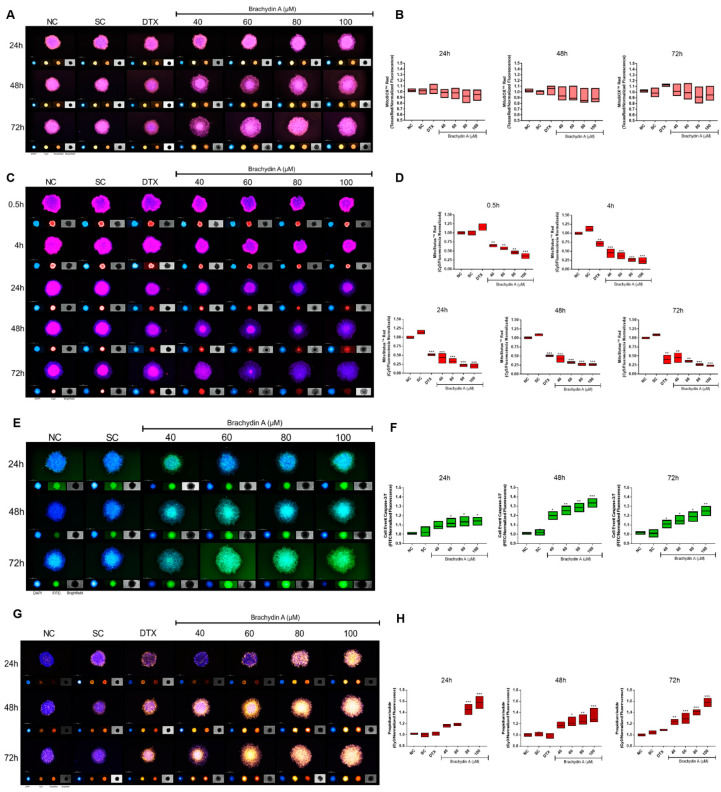
(**A**,**C**,**E**,**G**) Representative images of the effects of Brachydin A (BrA) and controls on mitochondrial ROS (MitoSOX Red), mitochondrial membrane potential (MitoStatus), apoptosis (Caspase 3/7), and necrosis (PI) in DU145 spheroids, respectively. The photomicrographs were acquired using the In-Cell Analyzer 2200 platform, using a 4× objective, in DAPI (Hoechst 33342), Cy5 (MitoStatus), FITC (CellEvent Caspase 3/7), and Cy3 (PI) channels. Scale: 200 µm. (**B**,**D**,**F**,**H**) Fluorescence intensity for MitoSOX Red (TexasRed), MitoStatus Red (Cy5), Caspase 3/7 (FITC), and PI (Cy3) in DU145 spheroids after treatment with BrA and respective controls. Data were normalized to the mean fluorescence intensity of the NC group at each time/point. The bars represent the mean ± standard deviation of four (*n* = 4) spheroids/replicates from three biological experiments (*n* = 3). * Values statistically different from the NC group (* *p* < 0.05; ** *p* < 0.01; *** *p* < 0.001; ANOVA followed by Dunnett’s post-test). NC: negative control (RPMI 1640); SC: solvent control (1% DMSO); DTX: Docetaxel (50 µM; positive control).

**Figure 7 pharmaceutics-14-00963-f007:**
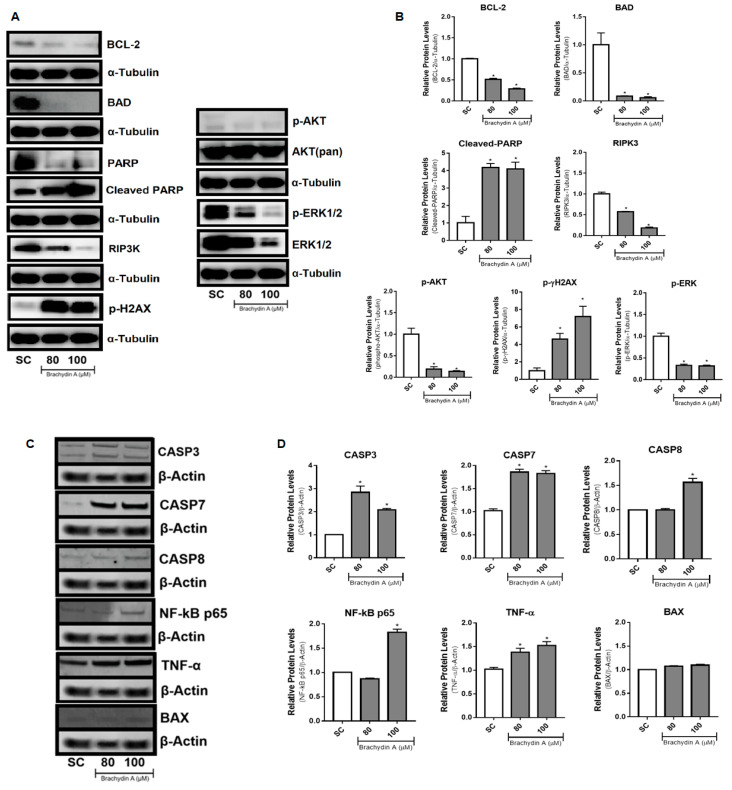
Western blotting analysis of protein expression and densitometric quantifications in DU145 spheroids treated with BrA for 24 h (**A**,**B**) and 72 h (**C**,**D**). Protein expression was normalized with α-Tubulin (24 h) and β-Actin (72 h) from the SC group. All bars are presented as mean ± standard deviation of experiments performed with ninety-six (*n* = 96) spheroids/replicates and two biological experiments (*n* = 2). * Statistically different values from the SC group (*p* < 0.05; ANOVA followed by Dunnett’s post-test). SC: solvent control (1% DMSO).

**Figure 8 pharmaceutics-14-00963-f008:**
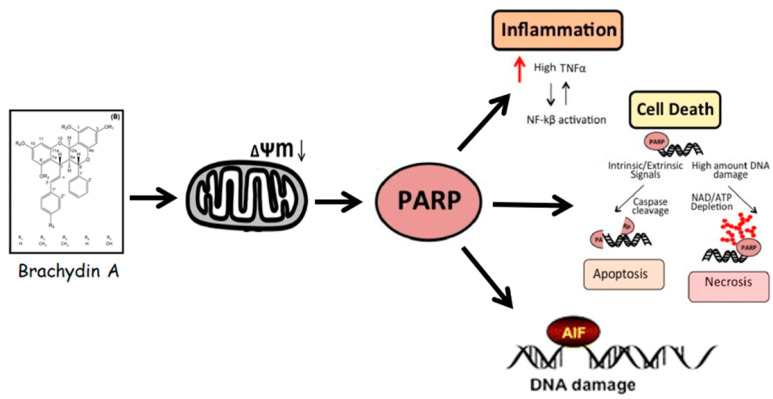
A proposed model of Brachydin A (BrA) effects in mPCa (DU145) tumor spheroids. BrA initially causes mitochondrial dysfunction and DNA fragmentation by PARP overactivation. Subsequently, cleaved-PARP promotes apoptosis/necrosis mixed-effects, DNA damage, and inflammatory response, suggesting that parthanatos cell death mediates the antiproliferative, cytotoxic, and antimetastatic properties observed in DU145 spheroids.

**Table 1 pharmaceutics-14-00963-t001:** Cell viability (%) of DU-145 spheroids after 24, 48, and 72 h of exposure with Brachydin A (BrA) and their respective controls, as assessed using the acid phosphatase assay ^1^.

			Brachydin A (µM)					
Exposure Time (h)	NC	SC	10	20	40	60	80	100
24	100.0 ± 2.0	102.1 ± 2.3	104.9 ± 1.6	93.9 ± 6.1	104.7 ± 5.2	97.9 ± 7.8	94.7 ± 5.8	95.6 ± 2.1
48	100.0 ± 1.0	104.5 ± 3.0	105.2 ± 4.9	107.6 ± 3.7	96.8 ± 9.4	87.4 ± 1.6 *	75.6 ± 7.7 **	74.2 ± 6.1 **
72	100.0 ± 1.7	98.6 ± 10.9	102.2 ± 6.5	103.3 ± 4.3	94.4 ± 8.8	89.0 ± 4.1 *	78.3 ± 6.2 **	79.7 ± 6.6 **

^1^ All values are presented as the mean ± standard deviation (X ± SD) of six spheroids/replicates (*n* = 6) in three biological experiments (*n* = 3). Values statistically different from NC at the respective time point (day) (* *p* < 0.05; ** *p* < 0.01; ANOVA followed by Dunnett’s post-test). NC: negative control (RPMI 1640); SC: solvent control (1% DMSO).

**Table 2 pharmaceutics-14-00963-t002:** Cell viability (%) of DU-145 spheroids after 24, 48, 72, and 168 h of exposure with Brachydin A (BrA) and their respective controls, as assessed using the resazurin assay ^1^.

				Brachydin A (µM)					
Exposure Time (h)	NC	SC	DTX	10	20	40	60	80	100
24	100.3 ± 1.0	98.6 ± 0.6	90.7 ± 1.9	95.6 ± 0.5	96.7 ± 1.3	93.4 ± 0.5	91.6 ± 1.1	93.1 ± 1.0	86.2 ± 0.6 *
48	100.7 ± 1.1	98.4 ± 2.4	93.4 ± 4.8 *	97.2 ± 1.2	98.0 ± 0.4	96.7 ± 1.7	90.7 ± 2.8 **	85.3 ± 4.1 **	81.2 ± 2.7 **
72	100.0 ± 1.7	97.2 ± 0.7	90.5 ± 0.5 *	99.2 ± 1.1	97.5 ± 0.8	95.2 ± 1.6	89.6 ± 1.1 *	75.1 ± 2.6 *	64.7 ± 3.1 *
168	100.0 ± 2.0	91.7 ± 1.3	24.7 ± 1.6 ***	86.7 ± 2.4 **	83.4 ± 9.3 ***	33.6 ± 5.7 ***	20.6 ± 2.7 ***	11.7 ± 1.2 ***	11.3 ± 0.5 ***

^1^ All values are presented as the mean ± standard deviation (X ± SD) of six spheroids/replicates (*n* = 6) in three biological experiments (*n* = 3). * Values statistically different from NC at the respective time point (day) (* *p* < 0.05; ** *p* < 0.01; *** *p* < 0.001; ANOVA followed by Dunnett’s post-test). NC: negative control (RPMI 1640); SC: solvent control (1% DMSO); DTX: Docetaxel (50 μM; positive control).

## Data Availability

Not applicable.

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
