# Peer review of "The Antitumoral/Antimetastatic Action of the Flavonoid Brachydin A in Metastatic Prostate Tumor Spheroids In Vitro Is Mediated by (Parthanatos) PARP-Related Cell Death"

_pharmaceutics, 2022, doi:10.3390/pharmaceutics14050963_

Round 1

Reviewer 1 Report

Comments:

The submitted manuscript by Lusânia and coworkers investigated the antiproliferative/antimetastatic effects of Brachydin A(BrA) and the molecular mechanisms underlying its action in mPCa spheroids in vitro. Observed altered spheroid morphology/volume and suppressed cell migration, and tumor invasive  60-100 μM concentration of BrA. 
Overall the manuscript was organized very well and scientifically strong. Therefore the current version of the manuscript is suitable for Pharmaceutics.

Author Response

Comments to the Authors and Responses

Reviewer 1:

#C1: The submitted manuscript by Lusânia and coworkers investigated the antiproliferative/antimetastatic effects of Brachydin A(BrA) and the molecular mechanisms underlying its action in mPCa spheroids in vitro. Observed altered spheroid morphology/volume and suppressed cell migration, and tumor invasive 60-100 μM concentration of BrA. Overall, the manuscript was organized very well and scientifically strong. Therefore, the current version of the manuscript is suitable for Pharmaceutics.

#R1: The authors are grateful to #reviwer 1 for his comments and for considering our manuscript to be published in Pharmaceutics.

Reviewer 2 Report

In the paper entitled “The antitumoral/antimetastatic action of the flavonoid Brachydin A in metastatic prostate tumor spheroids in vitro is mediated by (parthanatos) PARP-related cell death”, the authors investigated the antiproliferative and antimetastatic effects of Brachydin A, along with the molecular mechanisms based on its action in mPCa spheroids (DU145) in vitro. It was found that, under the concentrations of 60 ~ 100 μM, Brachydin A displayed cytotoxic activity by changing spheroid morphology and volume, and suppressing cell migration and tumor invasiveness. The mechanism of cell death caused by Brachydin A was that it could reduce mitochondrial membrane potential and increase apoptosis and necrosis markers. This work revealed that PARP-mediated cell death (parthanatos) is a potential mechanism of action of Brachydin A. which would provide a theoretical basis for the preclinical studies of Brachydin A as a potential drug against metastatic prostate cancer. I recommend this manuscript be accepted after minor revision.

  • When assaying the cytotoxic activity of Brachydin A against DU145 prostate tumor spheroids, the authors employed two assay methods, phosphatase and resazurin assays. What was the difference of the two methods? And please explain the reasons about why to choose both of them rather than one of the assay methods.
  • The manuscript did not mention the cytotoxic activity of Brachydin A against human normal cells, which was an important index for the evaluation of new drugs.
  • What was the interactions of Brachydin A in the active pocket of the possibly biological targets?
  • Please supplement the chemical structure of Brachydin A in the “Introduction” section.

Author Response

Comments to the Authors and Responses 

Reviewer 2:

All modifications made in the original manuscript are marked in red in the revised manuscript.

#C1: In the paper entitled “The antitumoral/antimetastatic action of the flavonoid Brachydin A in metastatic prostate tumor spheroids in vitro is mediated by (parthanatos) PARP-related cell death”, the authors investigated the antiproliferative and antimetastatic effects of Brachydin A, along with the molecular mechanisms based on its action in mPCa spheroids (DU145) in vitro. It was found that, under the concentrations of 60 ~ 100 μM, Brachydin A displayed cytotoxic activity by changing spheroid morphology and volume and suppressing cell migration and tumor invasiveness. The mechanism of cell death caused by Brachydin A was that it could reduce mitochondrial membrane potential and increase apoptosis and necrosis markers. This work revealed that PARP-mediated cell death (parthanatos) is a potential mechanism of action of Brachydin A. which would provide a theoretical basis for the preclinical studies of Brachydin A as a potential drug against metastatic prostate cancer. I recommend this manuscript be accepted after minor revision.

#R1: The authors are grateful to #reviwer 1 for his comments and for considering our manuscript to be published in Pharmaceutics.

#C2: When assaying the cytotoxic activity of Brachydin A against DU145 prostate tumor spheroids, the authors employed two assay methods, phosphatase and resazurin assays. What was the difference of the two methods? And please explain the reasons about why to choose both of them rather than one of the assay methods.

#R2: We understand your inquiry, and we believe that resazurin reduction and phosphatase activity produced a richer picture of the tumor spheroid condition and provided more consistent data on the cytotoxicity of Brachydin A (BrA). The acid phosphatase assay (APH) is a colorimetric cell viability method based on the dephosphorylation of the p-nitrophenyl phosphate substrate to the p-nitrophenol product by intracellular acid phosphatases enzymes of viable cells [1,2]. The acid phosphatase enzyme has a role in cell death, apoptosis, and autophagy. The extensive cell kills induced at high BrA concentrations could be triggering a decrease in acid phosphatase activity in DU145 cells. The resazurin (alamarblue®) assay is a fluorescent/colorimetric method based on the metabolic activity of live cells (within their cytoplasm and mitochondria) that reduces the resazurin dye to resorufin product by dehydrogenase enzymes [3,4]. The APH and resazurin assays were chosen because they are the most acceptable colorimetric methods used for cell viability measurement using 3D tumor spheroids [5]. So, to clarify in the manuscript text, we inserted a brief sentence about the principle of each method in the M&M section.  

Page 3 – Line 116/117: “to quantify viable cells from the cytosolic acid phosphatase enzyme activity”

Page 3 – Line 133/134: “to assess cell viability cells from the resazurin dye reduction performed by dehydrogenase enzymes”

References cited:

[1] Friedrich, J.; Seidel, C.; Ebner, R.; Kunz-Schughart, L.A. Spheroid-based drug screen: considerations and practical approach. Nat Protoc 2009, 4, 309-324.

[2] Friedrich, J., Eder, W., Castaneda, J., Doss, M., Huber, E., Ebner, R., Kunz-Schughart, L.A. A reliable tool to determine cell viability in complex 3-d culture: the acid phos-phatase assay. J. Biomol. Screen 2009, 12, 925–937.

[3] Riss, T.L.; Moravec, R.A.; Niles, A.L. Cytotoxicity testing: measuring viable cells, dead cells, and detecting mechanism of cell death. Methods Mol Biol 2011, 740, 103-114, doi:10.1007/978-1-61779-108-6_12.

[4] Walzl, A.; Unger, C.; Kramer, N.; Unterleuthner, D.; Scherzer, M.; Hengstschläger, M.; Dolznig, H.. The resazurin reduction assay can distinguish cytotoxic from cytostatic compounds in spheroid screening assays. J Biomol Screen 2014, 19, 7, 1047-1059.

[5] Pinto, B.; Henriques, A. C.; Silva, P.; Bousbaa, H. Three-dimensional spheroids as in vitro preclinical models for cancer research. Pharmaceutics 2020, 12, 12, 1186.

#C3: The manuscript did not mention the cytotoxic activity of Brachydin A against human normal cells, which was an important index for the evaluation of new drugs.   

#R3: The authors are grateful for the reviewer's comment. However, in the discussion section (Page 16 - Line 491-495), we mentioned the previous study published by our research group that showed the IC50 values obtained with non-tumor prostate epithelial (PNT2) cells (IC50 = 77.7 μM) in resazurin assay after Brachydin A (BrA) treatments. We also cited the higher selectivity potential (7.3 times) of the BrA flavonoid to induce cytotoxic effects on metastatic prostate DU145 cells (IC50 = 10.7 μM) cultured at 2D monolayers [1].

References cited:

[1] Oliveira, L. C. B.; Nunes, H. L.; Ribeiro, D. L.; Nascimento, J. R.; da Rocha, C. Q;, Colus, I. M. S.; Serpeloni, J. M.. Aglycone flavonoid brachydin A shows selective cytotoxicity and antitumoral activity in human metastatic prostate (DU145) cancer cells. Cytotechnology 2021, 73, 6, 761-774.

#C4: What was the interactions of Brachydin A in the active pocket of the possibly biological targets?

#R4: Thanks for the reviewer’s inquiry. To provide an accurate response, we believe that it would be necessary to carry out molecular docking predictions. Unfortunately, our research group currently does not have the necessary experience to carry out this type of analysis. Besides, our main objective was to evaluate the antitumoral effects of BrA flavonoid using a robust metastatic PCa model in vitro and provide a biological mechanism of action for the first time in literature. In the discussion section, we hypothesize that BrA possibly promotes metabolic stress and activates PARP-related cell death [1]. Besides being involved in DNA repair process, the PARPs enzymes are involved in metabolic regulation. Eventually, cytotoxic substances can promote intracellular NAD+ depletion and cause mitochondrial membrane depolarization. The release of the mitochondrial enzymes (i.e., AIF/MIF) activates PARPs (especially PARP1) and the entire downstream molecular cascade related to parthanatos [2-4]. Our results showed that BrA flavonoid causes Δψm depolarization after 30 min- treatment, and the many downstream consequences related to PARP-overactivation were observed on DU145 spheroids at phenotypic/molecular levels (flow cytometry, HCS, and western blotting).

References cited:

[1] Ying W. H.; Alano C. C.; Garnier P.; Swanson R. A. NAD(+) as a metabolic link between DNA damage and cell death. J Neurosci Res 2005, 79, 216–23.

[2] Bai, P.; Cantó, C. The role of PARP-1 and PARP-2 enzymes in metabolic regulation and disease. Cell Metabol 2012, 16, 3, 290-295.

[3] Morgan, M. J.; Liu, Z. G. 2013. Programmed cell death with a necrotic-like phenotype. BioMol Concepts 2013, 4, 3, 259-275.

[4] Zhou, Y.; Liu, L.; Tao, S.; Yao, Y.; Wang, Y.; Wei, Q.; Deng, Y. Parthanatos and its associated components: promising therapeutic targets for cancer. Pharmacol Res 2021, 163, 105299.

#C5: Please supplement the chemical structure of Brachydin A in the “Introduction” section.

#R5: The authors are grateful for the reviewer's suggestion. The authors agree with the suggestion and include the chemical structure of Brachydin A in Figure 1 in the introduction section (Page 2).
